# Utilisation of Skilled Birth Attendant in Low- and Middle-Income Countries: Trajectories and Key Sociodemographic Factors

**DOI:** 10.3390/ijerph182010722

**Published:** 2021-10-13

**Authors:** Tania Walker, Mulu Woldegiorgis, Jahar Bhowmik

**Affiliations:** 1Department of Health Science and Biostatistics, Swinburne University of Technology, Melbourne, VIC 3122, Australia; tcjwalker@gmail.com (T.W.); mwoldegiorgis@swin.edu.au (M.W.); 2Public Health Discipline, Burnet Institute, Melbourne, VIC 3004, Australia

**Keywords:** skilled birth attendants, low- and middle-income countries, maternal mortality ratio, sociodemographic, trajectory

## Abstract

Reducing the maternal mortality ratio (MMR) in low- and middle-income countries (LMICs) remains a huge challenge. Maternal mortality is mostly attributed to low coverage of maternal health services. This study investigated the trajectories and predictors of skilled birth attendant (SBA) service utilisation in LMIC over the past two decades. The data was sourced from standard demographic and health surveys which included four surveys on women with livebirth/s from selected countries from two regions with a pooled sample of 56,606 Indonesian and 63,924 Nigerian respondents. Generalised linear models with quasibinomial family of distributions were fitted to investigate the association between SBA utilisation and sociodemographic factors. Despite a significant improvement in the last two decades in both countries, the change was slower than hope for, and inconsistent. Women who received antenatal care were more likely to use an SBA service. SBA service utilisation was significantly more prevalent amongst literate women in Indonesia (AOR = 1.39, 95% CI: 1.24–1.54) and Nigeria (AOR = 1.41, 95% CI: 1.31–1.53) than their counterparts. The disparity based on geographic region and social factors remained significant over time. Given the significant disparities in SBA utilisation, there is a strong need to focus on community- and district-level interventions that aim at increasing SBA utilisation.

## 1. Introduction

Maternal mortality is unacceptably high in the low- and middle-income countries (LMICs) around the globe. Despite being on the global agenda for decades [1], the World Health Organisation (WHO) estimates suggest that 295,000 women still die each year [2] due to largely preventable complications related to pregnancy, childbirth, or during the postnatal period [3]. Roughly 94% of all maternal deaths occur in LMICs, primarily in Sub-Saharan Africa and Southern East Asia [2].

As part of the continued effort to reduce maternal mortality rate (MMR), the Sustainable Development Goals (SDGs) were launched at the United Nations General Meeting (2015b), with SDG 3.1 specifically outlining the reduction of the global MMR to less than 70 per 100,000 live births by 2030 [4]. However, current trends show that it is unlikely to achieve this target without further intervention [2]. Indonesia, for example, had an MMR of 177 per 100,000 live births (2017) and has managed to decrease their MMR by an average of 3.1% over the past 10 year (Indonesia Maternal Mortality Rate 2000–2021, 2021), whist Nigeria’s MMR rate in 2017, for example, was 917 per 100,000 live births and has been averaging a decrease of only 0.96% per year over the past 10 years (Nigeria Maternal Mortality Rate 2000–2021, 2021). Neither of these countries are likely to hit the SDG3.1 goal at this pace. There is a well-established connection between the increased utilization of maternal healthcare services (MHS) and decreasing maternal and child morbidity and mortality [2,5]. However, the use of these MHS interventions is limited in LMICs, and delivery outside health institutional settings or at least with the assistance of a skilled birth attendant (SBA) results in unsafe and unhygienic conditions [6,7]. This is particularly important given that SBA utilisation interphases with vital postnatal care (PNC) for the mother and child [5].

Examining the factors influencing MHS utilisation, particularly SBA service utilization, is a growing research focus area. A review of existing literature highlighted the vast nature of this topic area, though numerous studies covering the determinants of MHS utilisation were found to focus only on individual or family level impacts on MHS utilisation [8]. Individual social determinant factors such as the mother’s age [9], education [10], health service awareness [11], child birth order [9], the father’s education, and household wealth quintile [12] have all been found to contribute to low service utilization in LMICs. These studies are often restricted to single LMICs from Sub-Saharan Africa (SSA) or Asia, however [13,14,15,16,17], or are contextual to infrastructure (e.g., facilities) [18] or specific demographic factors (e.g., young mothers) [8], and historically they have largely ignored accounting for individual-, household-, community-, and district-level factors in tandem.

The need to look beyond individual-level factors when examining the health-seeking behaviours, however, is increasingly being recognised [19]. Additionally, formative factors such as education and literacy, often considered individual-level factors found to impact MHS utilisation, are heavily impacted by the communities and districts within which the individual lives, and often loop back in terms of elements such as the effectiveness of media exposure for the individual and the dissemination of information in their local community. MHS utilization has been found to be largely restricted by access to MHS [18], with some studies considering accessibility of sexual and reproductive health to be one of the biggest MHS utilisation challenges in low-income countries [20]. Researchers who have explored wider impacts of MHS utilisation found a strong association between MHS utilisation and a lack of healthcare-resourcing elements, rather than those factors directly relating to the mother. Such factors, which include limited media exposure for the dissemination of healthcare information [11,20], clinic availability [21], distance to facility [22], etc., affect not only the individual’s likelihood to use proper MHS, but that of the wider communities and districts they live in. In the limited places where such access issues are being facilitated, particularly rurally, it has proven to be a viable solution to educate mothers and reduce the MMR [17].

To date, trajectories of, and contributing factors to, MHS in the developing countries have not been adequately researched and taking these wider social determinants of MHS utilizations into account is vital to reduce the MMR in these countries and start them on track towards SDG 3.1. Whilst countries do have unique combinations of these wider social determinant factors, there has been little research looking at the same core set of factors regarding different LMICs across the same timeframe. This study, therefore, was interested in looking at the trajectories of MHS utilisation and its associated factors across two different continents, using one representative country from each.

Specifically, this study aimed to explore the trajectories of, and contributing factors to, SBA service utilisation during delivery in one Sub-Saharan Africa (SSA) and one Southern Asian (SA) country over the past two decades. It addressed the key determinants found to be most influential in each representative country for communalities and potential differences, using available demographic and health surveys (DHSs) in the last two decades.

## 2. Materials and Methods

### 2.1. Study Design

The data was sourced from standard DHS: DHSs are cross-sectional, standardised, and nationally representative population-based surveys that collect information on basic demographic and health topics and generally yield large sample sizes. The DHS applies a two-stage cluster sampling framework. Enumeration areas (EAs) from census data are the primary sampling units. In stage one, a stratified sample of EAs (the clusters) are selected using the probability-proportional-to-size method. In stage two, a predetermined number of households are selected from the EAs using an equal-probability systematic-sampling method. A household questionnaire is then administered in each household. Among others, each eligible woman aged 15 to 49 (the sampling units of interest) is then interviewed with an individual survey [23].

This study utilised the individual women’s data [24] from the standard DHS conducted in SSA and SA during the period 2000–2020. We selected one country from each region—Indonesia from SA, and Nigeria from SSA. The selection criteria included the following requirements: (1) its most recent dataset is no older than 4 years, (2) it must have at least four standard DHS available for analysis, and (3) the datasets used for comparison must fall roughly within the same window of time (e.g., 2–3 years). The two countries that best suited these criteria were Indonesia and Nigeria, with four datasets each within the past 20 years (Indonesia: 2002/2003, 2007, 2012, and 2017; Nigeria: 2003, 2008, 2013, and 2018). With primary data processing for each country having already been performed prior to the release of the deidentified datasets for public use, the chosen datasets were reviewed and screened for quality assurance and combined to form a master dataset for each country.

### 2.2. Participants and Samples

For each country surveyed, all women aged 15 to 49 years from each household were eligible to participate. For this study, inclusion was limited to women with at least one live birth within the past 5 years, and analysis focused on their most recent birth only. The Indonesian four DHSs had a combined total sample of 56,607 respondents, whilst the four Nigerian DHSs had a combined total of 63,924 respondents. The creation of internationally comparable datasets relating to the demographic and health characteristics of populations in developing countries is said to be one of the most significant contributions of the MEASURE DHS Program [25].

### 2.3. Variables

SBA service utilisation at birth was chosen as the outcome variable. The SBA outcome variable was binary: women who had accessed an SBA service during the delivery of their most recent pregnancy (yes), and those who had not (no), based on country definitions of SBA. For Indonesia, this includes doctors, obstetricians, nurses, midwifes, and village midwifes [26]. For Nigeria, this includes doctors, nurses/midwives, and auxiliary nurses/midwives [27]. The percentages of women who attended SBA services through these subcategories during the period 2000–2018 for each of the countries can be found in Appendix A. On the basis of the literature review and study objectives [9,10,11,12,13,14,15,16,17,20,28,29], fourteen explanatory variables were selected: age (15–29 vs. 30–49), residence (rural vs. urban), educational attainment of respondent and their husband/partner (defined as whether they or their partner had secondary or higher schooling (yes vs. no)), autonomy (based on primary or shared participation in all household decision-making (yes vs. no)), media exposure (defined as exposure to newspaper, radio, or television at least once a week (yes vs. no)), birth order (defined as whether most recent child was a first or second order birth (yes vs. no)), antenatal care (ANC) received at least one visit prior to birth (yes vs. no), wealth quintile (defined as whether household is within the middle or upper wealth quintiles (yes vs. no)), literacy (defined as the ability to read at least part sentences (yes vs. no)), and distance to HCF (defined as whether the distance is perceived as a problem (yes vs. no)).

Computed variables such as *literacy* and *autonomy* were created in line with DHS guidelines [30]. A custom variable was also developed to control for regional impacts, based on assessing the SBA utilisation in geographic subgroups (provinces/regions) for each country. States/regions were then assigned to quartiles, based on this initial SBA utilisation rate. The 2007–2008 datasets were used as the reference for allocating these regions to a quintile, given DHS geographic restructuring in Indonesia, with new regions added from 2007–2008 onwards for Indonesian DHS. Additionally, the year of survey, as a four-point Likert scale, was also included as a covariate to control for the general situation within in each of the countries, with the passing of time.

### 2.4. Statistical Analysis

DHS-identified sampling strata and clusters were used to create unique clusters and strata. Each country’s sampling weights were denormalised for pooled analysis. On the assumption that data was missing at random [31], case-wise deletion was used to deal with any missing values. Missing data found during the creation of custom variables (e.g., autonomy) was also dealt with in accordance with DHS guidelines for variable creation [30]. Initially, bivariate analyses were run on data to explore the association between the outcome and explanatory variables [32], their distributions, and their trajectories over the span of the datasets [33]. Bivariate analysis (chi-square test of significance) was used to examine the percentage of SBA differences for each predictor within the specific country.

Finally, generalised linear models (GLMs) with quasibinomial family of distributions (binary logistic regression models) were fitted to evaluate the association between SBA utilisation and sociodemographic factors [33]. GLMs were fitted, adjusting for cluster- and strata-wise effects, as well as sampling weights to generalise the findings across the regions. Statistical significance was set at 5%. Case-wise deletion was applied for missing values on the assumption that data was missing at random in fitting the GLMs. Data analysis was performed utilising SPSS version 26 (SPSS Inc., Chicago, IL, USA).

## 3. Results

Of the women of reproductive age who had at least one live birth within five years preceding each survey, about half of the Indonesian participants were aged between 15 and 29 (50.1%) and just over half lived in rural areas (53.2%). The majority of Nigerian participants were aged between 15 and 29 years old (52.1%) and had a greater rural representation (64.9%). In both countries, just over half of the women were from middle- or upper-class households (Table 1).

Attainment of secondary or higher education between Indonesian women and their husbands/partners was found to be similar, whereas partners of Nigerian participants were significantly more likely to have attained secondary or higher education (Table 1). On average, literacy rates were substantially higher in Indonesia, with 94.9% of women able to read part or whole sentences, and similarly exposed to media (84.9%). By comparison, only about half (47.4%) of women in Nigeria had frequent media exposure, whilst only four in ten (41.7%) women could read and therefore potentially benefit from written media exposure. This is mainly because literacy in Nigeria has dramatically decreased, whilst literacy in Indonesia has slowly increased over the past two decades (Table 1).

In general, Indonesian women’s most recent birth was more likely to be their first or second, whilst Nigerian women’s most recent birth was more likely to be a higher order birth. Twice as many Nigerian women reported instances where the birth interval between their most recent and preceding birth was less than 24 months, compared to Indonesia. Geographic SBA utilisation showed that approximately 14.3% of women in Indonesia fell within regions with the lowest SBA utilisation, on average, whilst 41.6% of Nigerian women live within such areas. Whilst the upper quartiles are similar, Indonesia has a broader distribution around quartiles 2 and 3 compared to Nigeria’s bottom-heavy SBA distribution (Table 1). Most Indonesian women had at least one ANC appointment during their pregnancy (94.9%), whilst two in three Nigeria women received ANC during their pregnancy (62.1%). Overall, four in five Indonesian women reported making use of an SBA service (80.2%). Comparatively speaking, two in five Nigerian women made use of SBA services (42.0%). Figure 1 shows a significant growth trend of SBA utilization in Indonesia. Nigeria similarly had relative growth in SBA utilisation over the two decades, albeit slower and less consistent. These results correspond with the decreasing mortality rates for both countries during the period 2000–2018 (Appendix A).

The bivariate association results presented in Table 2 show that the rates of SBA utilisation for Indonesian and Nigerian women were significantly higher among the participants aged 30 to 49, urban residents, had secondary/higher education qualifications, husbands/partners who had secondary/higher education qualifications, had autonomy within their household, were exposed to media, had a birth interval of more than 24 months, delivered their first or second order child, received ANC at least once during their pregnancy, were in the middle or upper wealth quintiles, were able to read, and did not consider the distance to their HCF to be a big problem (*p* < 0.05).

The results of the fitted logistic regression models (Table 3) showed that, for both countries, most of the sociodemographic factors were found to be significantly associated with SBA utilization, after controlling for potential confounders. Table 3 illustrates that older Indonesian women (30–49 years) had a 58.2% higher utilisation of SBA than those aged 15 to 29 years (AOR = 1.58, 95% CI: 1.47–1.70), whilst women aged 30 and over in Nigeria were 24.8% more likely to utilise SBA than those under 30 years (AOR= 1.25, 95% CI: 1.18–1.33). Furthermore, the odds of using an SBA increased by approximately 50% for both Indonesian (AOR = 1.55, 95% CI: 1.44–1.67) and Nigerian women (AOR = 1.50, 95% CI: 1.41–1.60) in urban areas, compared to their regional counterparts. Nigerian participants whose husband/partner had a secondary or higher education qualification were 37.6% more likely to make use of an SBA, whilst Indonesian women in similar situations were almost twice as likely. Variation inflation factors (VIFs) for these models can be seen in the Appendix A.

Greater autonomy within the household increased Indonesian women’s likelihood of SBA utilisation by approximately 13.8% (AOR = 1.14, 95% CI: 1.06–1.22), and regular media exposure delivered a 16.9% higher SBA utilisation compared to those with less media exposure (AOR = 1.17, 95% CI: 1.08–1.26). The most influential self-empowerment factor measured, however, was a secondary or higher education qualification—making SBA more than twice as likely (AOR = 2.24, 95% CI: 2.08–2.42) compared to those with a lower qualification. For Nigerian women, greater autonomy increased the likelihood of SBA utilisation to a greater extent than Indonesian women (AOR = 1.22, 95% CI: 1.15–1.3), whilst regular media exposure had less effect on SBA utilisation, compared to those with less media exposure (AOR = 1.13, 95% CI: 1.06–1.19). Although not as influential as in Indonesia, participants with secondary or higher education qualifications in Nigeria were 76.3% more likely to use an SBA than those with a primary qualification or lower (AOR = 1.76, 95% CI: 1.66–1.90).

As shown in Table 3, those who received ANC were 7.6 times more likely to use an SBA in Indonesia than those who did not receive ANC during their pregnancy, and, even more impactful, this was 11.4 times more likely in Nigeria. Inversely, first and second order births had a higher increased likelihood of SBA utilisation than lower order births in Indonesia than Nigeria. Results indicate that women from middle and upper wealth quintiles were more than twice as likely to utilise an SBA in Indonesia than those who were in the lower quintiles, and, similarly, 77.3% more likely in Nigeria. Indonesian participants who could read at least part sentences had a 39.2% higher utilisation of SBA than those who could not read at all (AOR = 1.39, 95% CI: 1.24–1.54), whilst the likelihood to use an SBA decreased by 37.6% if the women considered the distance to their HCF to be a big problem, in comparison to those who considered the distance a small or insignificant problem (AOR = 0.62, 95% CI: 0.57–0.68). Although the impact of literacy for Nigerian participants was almost exactly the same as that seen in Indonesia, problematic distances to HCFs in Nigeria only decreased SBA utilisation by 24.5%.

The participants’ province/state of residence also played an important part, as Indonesian women within Regional Quartile 2 had a 30.3% lower prevalence of using SBAs than those within Regional Quartile 1, whilst this was amplified for those in Regional Quartile 3 and Regional Quartile 4—who were three and five times as likely, respectively (Table 3). In Nigeria, this gap between first and fourth quartiles seemed even more augmented—with women from Regional Quartile 2 70.6% more likely to make use of SBAs than those within Regional Quartile 1, whilst those in Regional Quartile 3 and Regional Quartile 4 were 5.8 and 11 times as likely, respectively (Table 3).

Two separate binary logistic regression models by using first survey (2002–2003) and fourth survey (2017–2018) data for each country separately were also fitted to evaluate the changes that occurred from 2002–2003 to 2017–2018. The results obtained from the fitted models suggest that most of the sociodemographic factors remain significant covariates in both phases (2002–2003 and 2017–2018) of the surveys (Appendix A).

## 4. Discussion

This study evaluated the consistency and trajectory of SBA utilisation, as a proxy for maternal healthcare, using DHSs data from 2002 to 2018. Overall, SBA utilisation was found to vary both within and between countries. Trajectory results show that SBA utilization has increased, in absolute terms, for both Indonesia and Nigeria over this period, although Indonesian growth was more consistent and exponentially higher, and whilst Nigerian SBA use did increase over the study period, no utilisation growth was recorded between 2008 and 2013—and overall SBA utilisation remains low.

In line with existing literature [5,34,35,36], ANC (at least once during the pregnancy) from a skilled provider was found to be the most influential driver of SBA utilisation in both countries. The attendance of ANC appointments has the potential to alert the mother to any potential complications, which may require the use of delivery care services, making the association between ANC and SBA critical [20]. ANC is also highly associated with postnatal check-ups, and therefore MHS as a whole [17,34].

This study found that SBA utilisation fluctuated greatly by geographic region or state over time. Regions that were allocated to the top quartile generally stayed ahead of the other geographic quartiles in this aspect. Whilst some SBA utilisation increases were observed by all quartiles, the disparity between quartiles has not sufficiently been addressed, as significant differences were still found between these groups over the two decades.

The gap between the regional SBA utilisation quartiles is also of concern, with the gap in SBA utilisation in Nigeria between RQ1 and RQ4 being 71%, compared to Indonesia’s 25% range. Given that two in five people live in areas assigned to RQ1 in Nigeria, this is a significant issue [37]. A triple tier of financial and organisation responsibility between the federal, state, and local governments and a skewed distribution of trained staff between urban/rural and states are believed to contribute to overall insufficient healthcare services delivery [37]. Increased service provisions and the availability of free, or significantly more affordable, healthcare will be paramount to combating low service utilisation.

Consistent with past studies, those living in urban areas and those who have partners who had attained higher education levels and/or have done so personally are significantly more likely to use SBA services during delivery [4,16,21]. A positive association between wealth and MHS utilization exists, which could be due to the greater financial prioritization of primary needs over healthcare needs by less-wealthy families [17]. This further reinforces the need for affordability of MHS provision and access to the HCF as a primary focus for policy and infrastructure developers.

In line with the findings of other studies, this study revealed that level of education is linked to increased ANC utilisation and HCF delivery, as better-educated individuals are more knowledgeable of health literacy, and as such are likely to seek higher quality health services [36,38,39].

The study results also showed increased utilisation in association with female empowerment aspects such as autonomy within their households and regular media exposure, as well as increased literacy rates consistent with past studies [35,40]. Media access, higher level of education, and literacy were also greater amongst wealthy families, amplifying its effect and widening the gap between these groups [40]. Educated partners similarly have the potential to promote better health knowledge and service utilisation [41]. In countries such as Nigeria, where female education is less prevalent, educated partners are very valuable, given their ability to convey health information to and from their partner. From a policy perspective, therefore, involving the partner in the MHS discussion, e.g., through ANC visits or media messaging, is very important.

This study had some limitations to be considered when interpreting the results. Firstly, as this study used cross-sectional data, no causal inferences can be made. Secondly, the DHS data also relies on recall and self-reporting—but given it relates to maternal healthcare events, it is reasonable to assume biases will be less likely than some other topics. Thirdly, the use of 2007–2008 as the reference point for regional quartile allocation based on SBA utilisation (due to the addition of new geographic regions in Indonesia, from 2007–2008 onwards for the Indonesian DHS) could potentially have impacted on which quartile a state/region was allocated, irrespective of its performance in 2002–2003. The SBA utilisation order for the regions remained relatively consistent, however, for both these time periods. Fourthly, in rare cases, some women might have been interviewed for two consecutive surveys, and this information was not available in the deidentified data used in this study. Lastly, the evaluation of maternal healthcare utilisation was in terms of SBA only.

## 5. Conclusions

This study not only evaluated the trajectory of SBA over the past two decades, but also determined the individual-, community-, and wider-level predictors of SBA utilisation as a proxy for maternal healthcare. Despite the positive growth trajectories in both selected countries in SSA and SA, some community- and district-level predictors, such as education and geographic areas, have been identified as pivotal in significantly influencing SBA utilisation. Addressing those disparities will be fundamental to accelerating these growth patterns to give each country the best chance of achieving SDG 3.1. The disproportional impact of these factors on MHS makes them great starting points for policy developers in both countries looking to improve SBA utilisation and, resultantly, MHS, improving their chances of ultimately decreasing their MMR and reaching the SDG3.1 milestone by 2030.

Intervention programs focusing on education and health literacy in vulnerable cohorts, such as uneducated mothers and rural residents, could accelerate progress. Increasing SBA utilisation with regional disparities requires partnership on all levels of government, the private sector, and local communities to address shortcomings in any healthcare and educational resourcing.

## Figures and Tables

**Figure 1 ijerph-18-10722-f001:**
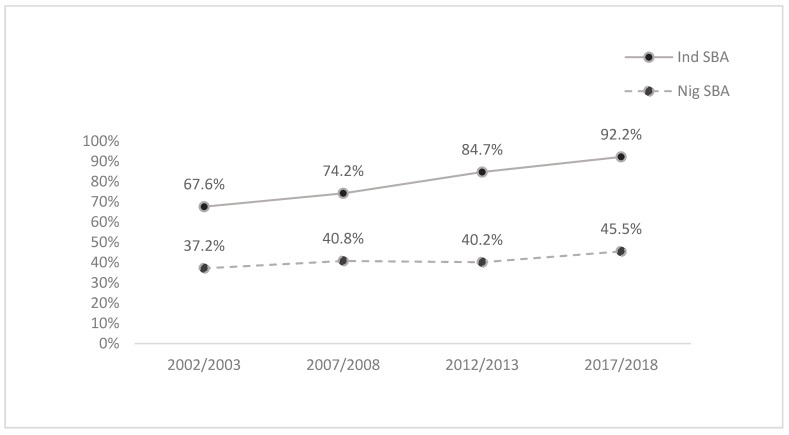
Trend of skilled birth attendant utilisation in Indonesia and Nigeria (2002–2018).

**Table 1 ijerph-18-10722-t001:** Distribution of the sociodemographic characteristics of the study participants.

Variables	Indonesia *n* (%)	Nigeria *n* (%)
Year of survey		
Period 1 (2002–2003)	12,760 (22.5)	3911 (6.1)
Period 2 (2007–2008)	14,043 (24.8)	17,635 (27.6)
Period 3 (2012–2013)	14,782 (26.1)	20,467 (32)
Period 4 (2017–2018)	15,021 (26.5)	21,911 (34.3)
Age		
15 to 29	28,384 (50.1)	33,331 (52.1)
30 to 49	28,222 (49.9)	30,594 (47.9)
Educational attainment of husband/partner		
Primary or lower	21,215 (37.9)	34,091 (56.1)
Secondary or higher	34,784 (62.1)	26,717 (43.9)
Has a say in household decision making		
No	15,356 (27.5)	42,424 (70)
Yes	40,426 (72.5)	18,197 (30)
Educational attainment of participant		
Primary or lower	22,122 (39.1)	41,676 (65.2)
Secondary or higher	34,484 (60.9)	22,249 (34.8)
Exposure to newspaper, radio and television is at least once a week		
No	8514 (15.1)	33,486 (52.6)
Yes	48,014 (84.9)	30,131 (47.4)
Birth interval between most recent and preceding birth less than 24 months		
No	33,115 (90.0)	41,970 (79.9)
Yes	3672 (10.0)	10,555 (20.1)
First or second order birth		
No	19,280 (34.1)	41,854 (65.5)
Yes	37,327 (65.9)	22,071 (34.5)
Received Antenatal care at least once during pregnancy		
No	2900 (5.1)	24,133 (37.9)
Yes	53,569 (94.9)	39,532 (62.1)
Wealth Index (Combined)		
Poor/Poorer	23,054 (40.7)	28,541 (44.6)
Middle/Upper	33,553 (59.3)	35,384 (55.4)
Place of residence		
Rural	30,106 (53.2)	41,459 (64.9)
Urban	26,501 (46.8)	22,465 (35.1)
Can read part or whole sentences		
No	2849 (5.1)	37,023 (58.3)
Yes	53,504 (94.9)	26,517 (41.7)
Had Skilled birth attendant at most recent live birth		
No	11,207 (19.8)	36,964 (58.0)
Yes	45,273 (80.2)	26,760 (42.0)
Perception of the distance to healthcare facility		
Small or no problem	49,270 (87.2)	43,390 (68.1)
Big problem	7249 (12.8)	20,365 (31.9)
Geographic Quartiles (based on 2007/2008 SBA ratings)		
Regional Quartile 1	8064 (14.3)	26,580 (41.6)
Regional Quartile 2	18,421 (32.6)	12,351 (19.3)
Regional Quartile 3	20,322 (35.9)	12,497 (19.5)
Regional Quartile 4	9765 (17.3)	12,496 (19.5)

**Table 2 ijerph-18-10722-t002:** Bivariate associations (using chi-square tests) between skilled birth attendance and sociodemographic factors for both Indonesia and Nigeria, 2002 to 2018.

	Skilled Birth Attendance					
	Indonesia			Nigeria		
	No (%)	Yes (%)	*p*-Value	No (%)	Yes (%)	*p*-Value
Year of survey						
2002/2003	4130 (32.4)	8600 (67.6)	<0.001	2451 (62.8)	1449 (37.2)	<0.001
2007/2008	3624 (25.8)	10,419 (74.2)		10,380 (59.2)	7157 (40.8)	
2012/2013	2256 (15.3)	12,466 (84.7)		12,195 (59.8)	8181 (40.2)	
2017/2018	1198 (8.0)	13,788 (92.0)		11,938 (54.5)	9973 (45.5)	
Age						
15 to 29 years.	5968 (21.1)	22,369 (78.9)	<0.001	20,127 (60.6)	13,099 (39.4)	<0.001
30 to 49 years.	5239 (18.6)	22,904 (81.4)		16,836 (55.2)	13,661 (44.8)	
Educational attainment of husband/partner						
Primary or lower	7678 (36.3)	13,489 (63.7)	<0.001	26,326 (77.5)	7623 (22.5)	<0.001
Secondary or higher	3432 (9.9)	31,278 (90.1)		9076 (34.0)	17,593 (66.0)	
Has a say in household decision making						
No	3541 (23.1)	11,788 (76.9)	<0.001	28,862 (68.3)	13,408 (31.7)	<0.001
Yes	7553 (18.7)	32,784 (81.3)		6582 (36.3)	11,575 (63.7)	
Educational attainment of participant						
Primary or lower	8280 (37.5)	13,778 (62.5)	<0.001	31,846 (76.7)	9663 (23.3)	<0.001
Secondary or higher	2928 (8.5)	31,494 (91.5)		5117 (23.0)	17,097 (77.0)	
Exposure to newspaper, radio and television is at least once a week						
No	3290 (38.8)	5200 (61.2)	<0.001	24,349 (73.0)	9012 (27.0)	<0.001
Yes	7895 (16.5)	40,016 (83.5)		12,412 (41.3)	17,646 (58.7)	
Birth interval between most recent and preceding birth less than 24 months						
No	7273 (22.0)	25,759 (78.0)	<0.001	25,205 (60.2)	16,649 (39.8)	0.014
Yes	956 (26.1)	2701 (73.9)		6462 (61.5)	4039 (38.5)	
First or second order birth						
No	5248 (27.3)	13,966 (72.7)	<0.001	26,168 (62.7)	15,538 (37.3)	<0.001
Yes	5960 (16.0)	31,307 (84.0)		10,796 (49.0)	11,223 (51.0)	
Received Antenatal care at least once during pregnancy						
No	2242 (77.3)	658 (22.7)	<0.001	22,460 (93.4)	1592 (6.6)	<0.001
Yes	8957 (16.7)	44,603 (83.3)		14,372 (36.4)	25,101 (63.6)	
Wealth Index (Combined)						
Poor/Poorer	7984 (34.7)	15,009 (65.3)	<0.001	24,183 (85.0)	4252 (15.0)	<0.001
Middle/Upper	3224 (9.6)	30,264 (90.4)		12,780 (36.2)	22,508 (63.8)	
Place of residence						
Rural	8495 (28.3)	21,548 (71.7)	<0.001	29,917 (72.40)	11,390 (27.60)	<0.001
Urban	2713 (10.3)	23,725 (89.7)		7046 (31.4)	15,370 (68.6)	
Can read part or whole sentences						
No	1600 (56.4)	1238 (43.6)	<0.001	29,460 (79.9)	7416 (20.1)	<0.001
Yes	9533 (17.9)	43,856 (82.1)		7273 (27.5)	19,193 (72.5)	
Perception of the distance to healthcare facility						
Small or no problem	8445 (17.2)	40,721 (82.8)	<0.001	22,194 (51.3)	21,076 (48.7)	<0.001
Big problem	2752 (38.1)	4475 (61.9)		14,663 (72.3)	5629 (27.7)	
Geographic Quartiles (based on 2008 SBA ratings)						
Regional Quartile 1	2736 (34.0)	5311 (66.0)	<0.001	22,471 (84.9)	4005 (15.1)	<0.001
Regional Quartile 2	4703 (25.6)	13,664 (74.4)		8462 (68.8)	3834 (31.2)	
Regional Quartile 3	2902 (14.3)	17,395 (85.7)		4286 (34.3)	8194 (65.7)	
Regional Quartile 4	863 (8.9)	8872 (91.1)		1744 (14.0)	10,727 (86.0)	

**Table 3 ijerph-18-10722-t003:** Results of logistic regression models for SBA: predictors of SBA utilisation for both Indonesia and Nigeria.

	Indonesia (N = 56,607)		Nigeria (N = 63,924)	
Characteristic	Unadjusted OR (95% CI)	Adjusted OR (95% CI)	Unadjusted OR (95% CI)	Adjusted OR (95% CI)
Year of survey (Ref: Period 1)				
2007/2008	1.38 (1.31–1.46) ***	1.28 (1.18–1.38) ***	1.17 (1.09–1.25) ***	0.93 (0.82–1.05)
2012/2013	2.65 (2.5–2.81) ***	2.20 (2.02–2.4) ***	1.13 (1.06–1.22) ***	0.94 (0.83–1.05)
2017/2018	5.53 (5.16–5.93) ***	4.7 (4.26–5.18) ***	1.41 (1.32–1.52) ***	1.14 (1.02–1.28) *
Sociodemographic Factors				
Age (30–49 vs. 15–29)	1.17 (1.12–1.22) ***	1.58 (1.47–1.7) ***	1.25 (1.21–1.29) ***	1.25 (1.18–1.33) ***
Father attained secondary education or higher (yes vs. no)	5.19 (4.96–5.43) ***	1.85 (1.72–1.99) ***	6.69 (6.46–6.94) ***	1.38 (1.29–1.47) ***
Empowerment				
Autonomy (yes vs. no)	1.3 (1.25–1.36) ***	1.14 (1.06–1.22) ***	3.79 (3.65–3.93) ***	1.22 (1.15–1.3) ***
Secondary or higher education (yes vs. no)	6.46 (6.17–6.77) ***	2.24 (2.08–2.42) ***	11.01 (10.59–11.44) ***	1.76 (1.62–1.92) ***
Media exposure at least once a week (yes vs. no)	3.21 (3.05–3.37) ***	1.17 (1.08–1.26) ***	3.84 (3.72–3.97) ***	1.13 (1.06–1.19) ***
Family Planning				
Birth interval less than 24 months (yes vs. no)	0.8 (0.74–0.86) ***	0.98 (0.88–1.08)	0.95 (0.91–0.99) *	0.94 (0.88–1.01)
First or second birth order (yes vs. no)	1.97 (1.89–2.06) ***	1.34 (1.24–1.43) ***	1.75 (1.69–1.81) ***	1.15 (1.07–1.24) ***
At least one Antenatal appointment (yes vs. no)	16.97 (15.51–18.56) **	7.60 (6.71–8.61) ***	24.65 (23.33–26.04) ***	11.38 (10.55–12.27) ***
Economic Accessibility				
Wealth Quintile (upper middle vs. Poor)	4.99 (4.77–5.23) ***	2.21 (2.06–2.38) ***	10.02 (9.63–10.42) ***	1.77 (1.66–1.9) ***
Community Impacting Factors				
Residence (urban vs. Rural)	3.45 (3.29–3.61) ***	1.55 (1.44–1.67) ***	5.73 (5.53–5.94) ***	1.50 (1.41–1.6) ***
Literacy (can read part or whole sentences vs. not)	5.95 (5.50–6.43) ***	1.39 (1.24–1.54) ***	10.48 (10.10–10.88) ***	1.41 (1.31–1.53) **
Distance to HCF (big problem vs. small or no problem)	0.34 (0.32–0.36) ***	0.62 (0.57–0.68) ***	0.40 (0.39–0.42) **	0.76 (0.71–0.8) ***
Geographic SBA Distribution (Ref: Quartile 1)				
Regional Quartile 2	1.5 (1.41–1.58) ***	1.30 (1.20–1.42) ***	2.54 (2.42–2.67) ***	1.71 (1.59–1.84) ***
Regional Quartile 3	3.09 (2.91–3.28) ***	2.95 (2.69–3.23) ***	10.72 (10.2–11.27) ***	5.84 (5.41–6.3) ***
Regional Quartile 4	5.3 (4.87–5.76) ***	4.98 (4.42–5.61) ***	34.51(32.47–36.67) ***	10.92 (9.99–11.93) ***

Notes: * = Significant at 0.05; ** = Significant at 0.01; *** = Significant at 0.001.

## Data Availability

This article does not contain any studies with human participants performed by any of the authors. Data used in research was attained from the National Institute of Population Research and Training (NIPORT), and ICF 2020, funded by the United States Agency for International Development (USAID). All identification of the respondents was deidentified before publishing the data. Views expressed in this study do not necessarily reflect those of USAID, the US government, NIPORT, or data custodians. The secondary datasets of the current study are available at https://dhsprogram.com/methodology/survey/survey-display-536.cfm. Permission for this project was taken from the Demographic and Health Surveys (DHS) Program authority by the authors.

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
