# Peer review of "Utilisation of Skilled Birth Attendant in Low- and Middle-Income Countries: Trajectories and Key Sociodemographic Factors"

_ijerph, 2021, doi:10.3390/ijerph182010722_

Round 1
Reviewer 1 Report
Specific comments are follows:
Comment #1: Study title: ‘Utilisation of maternal health services in LMICs …’ – maternal health services includes a broad area of care/services for mother or babies during antenatal and post-natal or postpartum period (this also include utilisation of SBA as well). SBA is just a single item measure ‘service used at delivery time’– it does not represent overall ‘maternal health services’ – although author mentioned in limitation section about the wording in title specifically ‘maternal health services’ defined by ‘SBA’– study limitation should not discuss the wording in title – in my suggestion author should change the title in line with what they measured in the study i.e. title can be ‘Utilisation of skilled birth attendant (SBA) in low- and middle-in-come countries (LMICs): trajectories and key socio-demo-graphic factors’.
Comment #2. This study utilised the individual women’s data from the standard DHS conducted during the period of 2000-2020. For the analysis purpose this study used combined data of 4 surveys which conducted over the 20 years of period. This duration of 20 years is too long to combined – over the 20 years health services improved quite a lot in most countries – so did the confidence level of general population to their national health services – also socioeconomic, cultural and political condition has changed as well. Fig1 shows that % of SBA utilisation in Indonesia has risen from 64% in 2002 to 92% (did not use SBA dropped: 36% to 8%). Utilisation of SBA in Indonesia for specific 4 surveys were: 67% (2002-03); 74% (2007-08); 85% (2012-13) and 92% (2017-18) respectively - so combining information from 4 surveys specially for Indonesia are not statistically justifiable – due to combining data overall rates for Indonesia is 75.8% - which is very close to 2007-08 rates i.e. rates for 10 years before the 4th survey data. Over the 20 years socio-demographic characteristics e.g. female literacy and employment has improved in LMICs. Although authors adjusted analysis for survey period – however all the predictors variable represent average of 4 surveys due to use of combined sample data. I would like to suggest in addition to current combined data analysis if the authors can do at least two separate models by using 1st survey (2002-03) and 4th survey (2017-18) data for each country separately – that will help policy makers and researchers to look at real trend –i.e. variations by socio-demographic factors remain same or changed over the year – that will help policy makers to identify what are the barriers in utilisation of SBA.
Comment #3. The only outcome factor in this study SBA service utilisation at birth was defined as the dichotomous outcome variable (used SBA, yes or no): women who had accessed an SBA service on at least one pregnancy considered as ‘utilised SBA (yes). This might have overestimated the % of SBA. For women with multiple pregnancies if the SBA utilisation for recent pregnancy/delivery is not known author can mentioned that as limitation section. Otherwise author should state the % of SBA rate for recent pregnancy/delivery.
Comment #4. ‘Antenatal care (ANC)’ defined as ‘made at least one visit prior to birth’ – it is not clear is this for recent pregnancy or at least one visit in any of their previous pregnancies. If the data for recent pregnancy are not available author should mentioned that as limitation. Otherwise author should state the ANC rate for recent pregnancy.
Comment # 5. In Table 2 authors used p-values – what kind of statistical test has been used in bivariate analysis are not mentioned in Statistical analysis section or in Table 2. The author should say either in methods or in Table 2 about the statistical test e.g. Chi-square test of significance was used to examine the % of SBA differences for each predictor within the specific country.
Comment #6. One of the main argument of this study was maternal mortality ratio (MMR) is unacceptably high in the LMICs – utilisation of SBA can improve that. The author did not present the MMR for the two selected LMICs (Indonesia and Nigeria) for this study. Author can find MMR for this two countries and from published sources and include in introduction which show the MMR situation of this two LMICs– also author can use a trend of MMR for last 20 years – that will be indirect proof that how MMR are associated with utilisation of SBA (i.e. MMR declined during past 20 years Indonesia as because of SBA rate increased).
Comment #7. For Indonesia SBA includes doctors, obstetricians, nurses, midwifes, and village midwifes; for Nigeria, SBA includes doctors, nurses/midwives, and auxiliary nurses/midwives. Is there any separate data available specific to each of SBA category i.e. % of women used doctors, obstetricians, nurses, midwifes, village midwifes’ during their delivery. If available, it’s better use as descriptive table/statistics for each of the DHS for Indonesia and Nigeria separately. That will be useful policy makers/researchers to take initiatives to improve utilisation of SBA and as well as improvement towards overall maternal health services.
Comment # 7. In absence of unique identification number for women participated in each DHS – there is a possibility that some women might have been interviewed two consecutive surveys – authors can indicate that as limitation. Alternatively, if possible authors can extract and identify how many are attended in more than surveys.
Author Response
Thank you for providing us the opportunity to revise our manuscript entitled, “Utilisation of maternal health services in low- and middle-income countries: trajectories and key socio-demographic factors”. The authors wish to thank the reviewers for the constructive feedback they have provided on our submission (ID:ijerph-1371000)). This feedback has enabled us to produce a stronger manuscript. We have systematically addressed the reviewers’ comments in our attached responses and incorporated related changes into the original submission through a major revision using track change mode as suggested.

Reviewer 2 Report
My only concern regards the inclusion if the variable "birth interval between most recent and preceding birth less than 24 months" in the multivariate analysis which limits the target population to only mothers with at least two live births. As it has been shown in Table 1, about 20.000 respondent from Indonesia and about 14.000 from Nigeria are being excluded from the model. There is no mention of this in the manuscript.
Below a minor revision:
- Figure 1: It is not clear which trend belong to Nigeria and Indonesia, respectively. Please adjust the legend and explain why in the lower trend a there is a continuous line.
- p 3, row 139: "States/ regions were then assigned to quantiles,...": from the analysis they seem to be assigned to quartiles. If so, use the term "quartiles".
- Table 2. Reporting p-values alone is not informative. ORs should be included as well. Moreover, check for vertical alignment of p-values.
- Table 3. Indonesia (N=63924) actually is 56607
- Table 3. Please mention why are using AORs and how these are calculated.
- Table 3. It is curious that there are not significant effects with p<0.001 given the large sample size (only 1 or 2 asterisks are reported). Please verify.
Author Response

(The authors gave the same response as above.)

Round 2
Reviewer 2 Report
I have found the current version of the manuscript enhanced, clearer and suitable for publication. Just check why there are two versions of Figure 1.
Author Response
Thank you for providing us the opportunity to revise our manuscript entitled, “Utilisation of maternal health services in low- and middle-income countries: trajectories and key socio-demographic factors”. The authors wish to thank you for the constructive feedback you have provided on our submission (ID:ijerph-1371000)). This feedback has enabled us to produce a stronger manuscript. We have systematically addressed the comments in our attached responses below and incorporated related changes into the original submission through a minor revision using track change mode as suggested. As instructed, our responses to the reviewers’ comments are provided below one by one. For convenience, the reviewers’ comments are highlighted in black and authors responses are highlighted in blue font. As suggested the supplementary materials are attached in a separate document and submitted separately. These supplementary tables have been deleted from the revised version of the manuscript.
Reviewer’s comment
Comments and Suggestions for Authors
I have found the current version of the manuscript enhanced, clearer and suitable for publication. Just check why there are two versions of Figure 1.
Authors’ response
We appreciate reviewer’s comment and thanks for reviewing revised version of our article. Actually, in the track change option we have deleted the older version of Table 1 and included updated version underneath the older version. Our apology for this confusion.